# Optimization of the Extraction from Spent Coffee Grounds Using the Desirability Approach

**DOI:** 10.3390/antiox9050370

**Published:** 2020-04-29

**Authors:** Maria Rosa Gigliobianco, Barbara Campisi, Dolores Vargas Peregrina, Roberta Censi, Gulzhan Khamitova, Simone Angeloni, Giovanni Caprioli, Marco Zannotti, Stefano Ferraro, Rita Giovannetti, Cristina Angeloni, Giulio Lupidi, Letizia Pruccoli, Andrea Tarozzi, Dario Voinovich, Piera Di Martino

**Affiliations:** 1School of Pharmacy, University of Camerino, 62032 Camerino, Italy; maria.gigliobianco@unicam.it (M.R.G.); dolores.vargas@unicam.it (D.V.P.); roberta.censi@unicam.it (R.C.); gulzhan.khamitova@unicam.it (G.K.); simone.angeloni@unicam.it (S.A.); giovanni.caprioli@unicam.it (G.C.); cristina.angeloni@unicam.it (C.A.); giulio.lupidi@unicam.it (G.L.); 2Department of Economic, Business, Matematic and Statistical Sciences, University of Trieste, 34127 Trieste, Italy; campisi@units.it; 3Recusol srl, Via del Bastione 16, 62032 Camerino, Italy; 4School of Science and Technology, Chemistry Division, University of Camerino, 62032 Camerino, Italy; marco.zannotti@unicam.it (M.Z.); stefano.ferraro@unicam.it (S.F.); rita.giovannetti@unicam.it (R.G.); 5Department for Life Quality Studies, University of Bologna, 40126 Bologna, Italy; letizia.pruccoli2@unibo.it (L.P.); andrea.tarozzi@unibo.it (A.T.); 6Department of Chemical and Pharmaceutical Sciences, University of Trieste, 34127 Trieste, Italy; vojnovic@units.it

**Keywords:** spent coffee grounds extract, circular economy, desirability approach, caffeine, trigonelline, nicotinic acid, total phenol content, antioxidant capacity, element content, keratinocyte HaCaT cells, cytotoxicity, reactive oxygen species

## Abstract

The purpose of this work was the optimization of the extraction from spent coffee grounds, specifically 100% Arabica coffee blends, using a desirability approach. Spent coffees were recovered after the preparation of the espresso coffee under the typical conditions used in coffee bars with a professional machine. Spent coffee was subjected to different extraction procedures in water: by changing the extraction temperature (60, 80, or 100 °C) and the solvent extraction volume (10, 20, 30 mL for 1 g of coffee) and by maintaining constant the extraction time (30 min). The ranges of the process parameters, as well as the solvent to be used, were established by running preliminary experiments not reported here. The variables of interest for the experimental screening design were the content of caffeine, trigonelline, and nicotinic acid, quantitatively determined from regression lines of standard solutions of known concentrations by a validated HPLC-VWD method. Since solvent extraction volumes and temperatures were revealed to be the most significant process variables, for the optimization of the extraction process, an approach based on Response Surface Methodology (RSM) was considered. In particular, a Box-Wilson Central Composite Design, commonly named central composite design (CCD), was used to find the optimal conditions of the extraction process. Moreover, the desirability approach was then applied to maximize the extraction efficiency by searching the optimal values (or at least the best compromise solution) for all three response variables simultaneously. Successively, the best extract, obtained in a volume of 20 mL of water at an extraction temperature of 80 °C, was analyzed for total phenol content (TPC) through the Folin–Ciocalteu assay, and the antioxidant capacities (AC) through the trolox equivalent (TE) antioxidant capacity (DPPH), ferric-ion reducing antioxidant parameter (FRAP), and radical cation scavenging activity and reducing power (ABTS). The TPC and the AC for spent coffee were high and comparable to the results obtained in previous similar studies. Then, the extract was evaluated by inductively coupled plasma mass spectrometry (ICP–MS), revealing that potassium was the most abundant element, followed by phosphorus, magnesium, calcium, sodium, and sulfur, while very low content in heavy metals was observed. Preliminary in vitro assays in keratinocyte HaCaT cells were carried out to assess the safety, in terms of cytotoxicity of spent coffee, and results showed that cell viability depends on the extract concentration: cell viability is unmodified up to a concentration of 0.3 mg/mL, over which it becomes cytotoxic for the cells. Spent coffee extract at 0.03 and 0.3 mg/mL showed the ability to reduce intracellular reactive oxygen species formation induced by hydrogen peroxide in HaCaT cells, suggesting its antioxidant activity at intracellular levels.

## 1. Introduction

The exploitation of agricultural waste in a circular economy has been quickly growing in the last few years [1]. In the same direction, the food industry tends to exploit different by-products to recycle ingredients to valorize and add economic value to waste [2]. Among all, spent coffee grounds (SCG) have been widely considered for their reuse [3]. The interest is related to coffee, the global consumption of which exceeded 9.3 billion kilograms in 2016, according to the International Coffee Organization, and which remains one of the most traded products in the world [4]. The main investigated applications in the reuse of spent coffee grounds are in the bio-energy for the production of bio-diesel, bio-ethanol, bio-oil and solid fuel, in the production of materials such as plastics, composites, and adsorbents, or finally in the extraction of bioactive compounds, such as phenol compounds, carotenoids, and fertilizers.

Concerning this latter application, the antioxidant activity of SCG can be ascribed to polyphenols, particularly chlorogenic acids such as caffeic acid [5,6], nicotinic acid [7], and trigonelline, which is converted to nicotinic acid under thermal treatment [8].

In previous studies, several extraction methods have been envisaged to optimize the recovery of active substances, in particular polyphenols. A summary of the results of these studies is provided in Table 1. In most cases, the efficacy of these methods has been evaluated through spectrophotometric methods and in particular through the total phenol content (TPC) and antioxidant capacities (AC), ferric-ion reducing antioxidant parameter (FRAP), radical cation scavenging activity and reducing power (ABTS) or trolox equivalent (TE) antioxidant capacity (DPPH), or more than one. The applied methods may differ in several parameters. For example, different extraction procedures have been considered, from the simplest solvent extraction [6,9] to ultrasound-assisted extractions [10,11,12], Soxhlet extraction [12,13], subcritical fluid extraction [12,14], or authohydrolysis [15]. Regardless of the method, one of the most important parameters is the extraction solvent, when concerned. Organic solvents like ethanol, hexane, and dichloromethane, particularly when combined with generally considered more efficient methods such as ultrasound or Soxhlet extractions, have been revealed as particularly efficient to provide extracts rich in antioxidants. Nonetheless, within the possibility to avoid non-green or toxic organic solvents, water and ethanol may be considered as solvents of first choice. Among them, ethanol or water–ethanol mixtures could offer very slightly better results [6]. Other frequently considered variables are extraction temperature and time, and in general, increasing time and temperatures offer advantages in terms of extraction efficiency, but not in terms of process costs.

By considering these premises, in the present study, we want to optimize an extraction method easily reproducible and industrially scalable. To this end, we propose an economic and simple apparatus and method (for example by using the lowest extraction temperature and time), able to avoid ethanol or other expensive, inflammable, or toxic extraction solvents. Differently from the previous study, the optimization will be based on the extraction of three different and specific molecules: caffeic acid, trigonelline and nicotinic acid. These molecules have been selected because they are among the most attractive active molecules contained in the SCG for cosmetic purposes. In particular, caffeic acid is very well known as an antioxidant with free radical scavenging properties [16], while trigonelline is beneficial for its antimicrobial properties, and thus it could aid in preventing the microbial contamination of the extracts and cosmetic formulations [17]. Niacinamide possesses a stabilizing effect on epidermal barrier function, reducing the trans epidermal water loss and thus favoring the skin hydration. Moreover, niacinamide stimulates the protein synthesis such as keratin, as well as ceramide synthesis, and favors the keratinocyte differentiation, with benefits on aged skin [18].

**Table 1 antioxidants-09-00370-t001:** Total Phenol Content (TPC) and antioxidant capacities (AC) (FRAP, ABTS, or DPPH) referred to different results reported in the scientific literature. All the values are converted into the same ones. When values are not provided, they may be not reported in the corresponding study, or not convertible into the same values.

Extraction Method	TPC mg GAE/g	FRAP TEAC (µmol TE/g)	ABTS TEAC (µmol TE/g)	DPPH TEAC (µmol TE/g)	References
2 g SCG in 100 mL pure water at 60 °C for 30 min	6.33–19.62 *	-	-		[6]
2 g SCG in 100 mL ethanol at 60 °C for 30 min	11.83–28.26 *	-	-		[6]
Ethanol extraction 40 mL solvent/g SCG, 70% EtOH, 50 °C, 2 h	17.09	-	-	-	[9]
Subcritical water extraction 179 °C, 36 min 14.1 g SCG/L	88.34	-	886.50 §	382.8 §	[14]
Subcritical water extraction (different temperatures, times, and ratio solid to liquid)	21–56 *	-	70–320 §*	50–220 §*	[14]
Autohydrolysis 15 mL water/g SCG, 200 °C, 50 min	40.36	-	125.69	112.47	[15]
Boiling water 10 g SCG/L, 10 min	5.66 ± 0.07	-	-	-	[19]
Ultrasound-assisted solid-liquid extraction 1g in 100 mL ethanol T = 30–50 °C Time= 5–45 min	33–36 *	-	-	-	[10]
Ultrasound Methanol/water 0.49-1.50 w/w Time = 9–112 min Time = 0.60–7.40 min	19–25 *	134–174 *		81–146 *	[11]
Soxhlet 2 g in 250 mL of hexane, 5 h	273.34 ± 34.17	-	-	148.40 ± 30.43	[13]
Supercritical fluid extraction Pressure = 100–300 T = 40–60 °C	17–28 *	-	38–54 *	-	[12]
Soxhlet 5 g in 150 mL hexane, 6 h	65–151 *	-	98–381 *	-	[12]
Ultrasounds 7 g in 210 mL Room temperature 6 h (dichloromethane, ethanol, or ethylacecate)	61–133.4 *	-	128–161 *	-	[12]

§ Original values reported in the corresponding study are converted here according to the same values used in this table for immediate comparison. * Data correspond to the ranges of the minimal and maximal values obtained according to the different conditions used during the corresponding study.

To this purpose, a plan of experiments was designed in order to recover the best extract with a few experiments [20]. Design of Experiments is very helpful to make more efficient and effective the scientific learning process and, in particular, Response Surface Methodology is useful for optimization purposes. The optimization strategy mainly used in scientific literature consists of the Response Surface Methodology through different experimental design approaches such as the factorial [9], central composite [11,14,21,22], and the Box-Behnken designs [10].

In our study, the Box-Wilson Central Composite Design (also known just with the acronym CCD), often applied in research investigations, was used to find the optimal conditions of the extraction process. With the aim to maximize the extraction efficiency for all the three response variables simultaneously, the desirability function approach was adopted. The approach here adopted involved also a reliability evaluation of the overall desirability function in order to identify a more robust optimal zone, including all the possible solutions respecting the given constraints with a probability ≥ 95%. The advantage of this method is to obtain a single function of multiple responses that measures an overall desirability for the process optimization. Typically, in many practical situations there are many response variables to be analysed and optimized. In these cases, it should be recommended not to optimize the single responses individually and independently of one another. The optimal solution found thanks to this desirabilty approach fulfils all the restrictions imposed on the individual functions. Morever, in this paper was proposed a method considering both the optimality and robustness of the solution.

The best extract was thus characterized for TAC and AC, mineral and metal content, and in vitro cytotoxicity and antioxidant activity assays. We envisage the application of this extract in the cosmetic field.

## 2. Materials and Methods

### 2.1. Materials and Standards

Pure standards of caffeine, trigonelline, and nicotinic acid were purchased from Sigma-Aldrich (Stenheim, Germany). Individual stock solutions of caffeine, trigonelline, and nicotinic acid, at a concentration of 1 mg/mL, were prepared by dissolving pure standard compounds in HPLC-grade methanol and storing them in glass-stoppered bottles at 4 °C. Afterward, standard working solutions at various concentrations were prepared daily by appropriate dilution of the stock solutions with methanol. HPLC-grade methanol was supplied by Sigma-Aldrich (Milano, Italy), and HPLC-grade formic acid (99%) was supplied by Merck (Darmstadt, Germany). Methanol as extraction solvent was also purchased from Sigma-Aldrich (Stenheim, Germany). The ultrapure water (resistivity > 8MΩ cm) was produced in house by Gradient Milli-Q^®^ (Millipore, Molsheim, France) and filtered with a 0.20 µm Sartolon polyamide filter (Sartorius Stedim Biotech, Göttingen, Germany).

The starting coffee consisted of a 100% grain Arabica from India and was provided by Illy Caffè S.p.A. (Trieste, Italy).

1,1-Diphenyl-2-picrylhydrazyl (DPPH), 2,4,6-Tris(2-pyridyl)-s-triazine (TPTZ), (±)-6-Hydroxy-2,5,7,8-tetramethylchromane-2-carboxylic acid (Trolox), 2,2′-Azino-bis(3-ethylbenzothiazoline-6-sulfonic acid) diammonium salt (98%TLC) (ABTS, gallic acid, sodium carbonate monohydrate ACS reagent, sodium acetate, and ethanol (ethanol absolute grade) were purchased from Sigma-Aldrich (Stenheim, Germany). Manganese (IV) oxidize activated (≥90%) and Folin–Ciocalteu’s phenol reagent were purchased from Fluka (Buchs, Switzerland). Anhydrous sodium acetate, Iron (III) anhydrous hydrochloride were purchased from J.T. Baker Analyzed (Center Valley, PA, USA), and sodium carbonate anhydrous was purchased from Carlo Erba (Milano, Italy). All solvents and reagents were of analytical grade.

### 2.2. Sample Extraction

Coffee beans, 100% Coffea arabica L., were milled in a coffee grinder (Mythos 1, Simonelli Group S.p.A., Belforte del Chienti, Italy). The espresso coffee was produced through a professional machine (VA388, Black Eagle, Victoria Arduino, Simonelli Group S.p.A., Belforte del Chienti, Italy) following these conditions: 7.5 g of the finely ground coffee for each cup; 9 bar of water pressure and 92 °C of water temperature; 25 ± 1 mL of product per cup; 25 ± 1 s of extraction.

Accurately weighted coffee grounds were desiccated in an oven (Heraeus, Hanau, Germania) at 50 °C for approximately 48 h to reach a constant weight. Extractions were carried out starting from 1 g of ground coffee exposed to water as extraction solvent for the same extraction time (30 min) in a water bath (Arex Heating mag. Stirrer, 230 V-50/60 Hz, code F20500413, Velp Scientifica Srl, Monza Brianza, Italia Velp Scientifica Srl, Monza Brianza, Italy) at different temperatures and in different solvent extraction volumes according to the experimental plan described in Section 2.5. Once extraction ended, samples were centrifuged at 4000 rpm (IEC CL 10 centrifuge, 230 V, 310 W, max speed 6500 rpm, model n°11210900 Thermo Scientific, Monza, Italy) to recover the limpid extract to be used for further studies. Extracts were freeze-dried at −50 °C and a pressure of 0.03 bar (FreeZone 1 Liter Benchtop Series 77400 freeze-dryer, LABCONCO, Kansans City, MO, USA) in 50 mL polyethylene vials with screw cap (BD Falcon ™, BD Biosciences, Bedford, MA, USA) and stored for future characterizations.

### 2.3. HPLC Analysis

The determination of extracts was provided by injecting 10 μL of sample in an HPLC-VWD (Agilent Technologies 1100 Series, Palo Alto, CA, USA) following a previously published paper (Caprioli et al., 2014) [23]. Briefly, the separation of the analytes was accomplished on a Gemini C18 110A analytical column (250 × 3 mm I.D., 5 mm) from Phenomenex (Chesire, UK). The mobile phase for HPLC-VWD analysis was water (A) containing 0.3% of formic acid and methanol (B), at a flow rate of 0.4 mL/min. The gradient program was: 0 min, 25% B; 0–10 min, 60% B; 10–15 min, 60% B; 15–20 min, 25% B; held at 25% until the end of the run at 25 min. Caffeine, trigonelline, and nicotinic acid were determined at 265 nm. Before HPLC analysis, all samples were filtered with Phenex™ RC 4 mm 0.45 μm syringeless filter (Phenomenex, Castelmaggiore, BO, Italy).

### 2.4. Method Validation

Calibration curves of caffeine, trigonelline, and nicotinic acid were constructed injecting in HPLC-VWD 10 µL of standard solutions at six different concentrations, i.e., 1, 5, 10, 50, 100, 250 mg L^−1^. Each calibration curve of the analyzed compounds showed a linear correlation coefficient (R2) greater than 0.9972. Repeatability of the chromatographic procedure was calculated at two concentration levels (20 and 50 mg L^−1^) through three replicate determinations on the same day and on three different days of a standard solution. Run-to-run precision Relative Standard deviation % (RSDs%) ranged from 0.23% to 0.91%; day-to-day precision ranged from 1.4% to 3.1%. The sensitivity was determined according to the limit of quantification (LOQ) and limit of detection (LOD). The LOD value was defined as the concentration at which the ratio between the peak height of each analyte (S) and the noise (N) is equal to 3. For the LOQ values, the S/N ratio must be equal to 10 [24]. The limit of detection (LOD) was 0.03 mg/kg for trigonelline and nicotinic acid and 0.06 mg/kg for caffeine; limit of quantification (LOQ) was 0.1 mg/kg for trigonelline and nicotinic acid and 0.2 mg/kg for caffeine. The method specificity was evaluated by measuring retention time stability for each molecule. The retention time stability was studied 3 times over a period of 3 days (*n* = 9) and expressed by RSD. % RSDs were in all cases lower than or equal to 1.17%.

### 2.5. Optimization of the Extraction Conditions

For this optimization study, a methodological approach based on experimental design and statistical analysis was chosen. The objective was to identify the best extraction conditions allowing simultaneously for the highest content of caffeine, trigonelline, and nicotinic acid. During preliminary essays, different variables were considered: the solvent (water, ethanol–water, glycerol–water), the solvent extraction volume, the extraction time, and the extraction temperature. Within this objective, two dependent variables were identified as significant: the solvent extraction volume and the extraction temperature. For each of these two variables, the corresponding ranges were also identified and are reported in Table 2. For the determination of the best extraction conditions, three independent variables—determined by HPLC-VWD analysis—were taken into account: the contents of caffeine (Y1), trigonelline (Y2), and nicotinic acid (Y3).

From the experimental domain, it was possible to build the experimental plan based on a Central Composite Design (CCD) consisting of a full-factorial two-level design (runs 1–4) with center points (runs 9 and 10) and the star or axial points (runs 5–8) given in Table 3. For this optimization study, to the classical CCD points, three test or check points were added (runs 11–13) to assess the adequacy of the postulated model (Figure 1).

The CCD is also known as a second-order design because it allows fitting and checking the second-degree polynomial model—including linear terms and squared terms for all factors, and products of all pairs of factors—given in Equation (1) [25]:(1)$$\eta =\beta_{0}+\overset{k}{\underset{i=1}{\sum }}\beta_{i}X_{i}+\overset{k}{\underset{i=1}{\sum }}\beta_{ii}X_{ii}^{2}+\overset{k-1}{\underset{i=1}{\sum }}\overset{k}{\underset{j=2}{\sum }}\beta_{ij}X_{i}X_{j}$$

### 2.6. Total Phenol Content Determination

The Total Phenol Content (TPC) of the optimized extract was determined according to the Folin–Ciocalteu spectrophotometric method [26] according to Zorzetto et al. [27,28].

### 2.7. Evaluation of Antioxidant Capacity

The antioxidant capacity (AC) of the extract was evaluated by measuring 1,1-diphenyl-2-picrylhydrazyl (DPPH•) radical scavenging activity, 2,2’-azino-bis (3-ethylbenzothiazoline-6-sulphonic acid) (ABTS•+) radical cation scavenging capacity, and Ferric Reducing Antioxidant Capacity (FRAP). Trolox (6-hydroxy-2,5,7,8-tetramethylchroman-2-carboxylic acid) was used as calibration standard. Values were expressed as IC50, defined as the concentration of the tested material required to cause a 50% decrease in initial DPPH, ABTS or iron concentration, and µmol Trolox equivalent/g of sample.

DPPH free radical scavenging activity was evaluated on a microplate analytical assay according to the previously published methods [29] with some modifications [28,30].

The FRAP values of extracts were determined according to a previously published method [31] with some modifications [28,32].

### 2.8. Determination of Metals and Minerals by Inductively Coupled Plasma Mass Spectrometry (ICP–MS)

Extract solution obtained by using 1 g di spent coffee in 20 mL of solvent was prepared and successively lyophilized. Ultrapure water obtained from a Millipore Milli-Q system (resistivity 18.2 MΩ cm) was used to prepare the solution for elements analysis. Specifically, 0.0157 g of lyophilized sample was placed in a Teflon digestion vessel with 1 mL of HNO_3_ (65%, Suprapur quality, Merck), 4 mL of H_2_O_2_, 0.5 mL of Ultrapure water, and 50 µL of a solution (2 mg/L) of Be, Ru, and Au as recovery standard. A microwave closed vessel system (Berghof Speedwave 4, Berghof, Eningen, Germany) was used for mineralization. The mineralized solution was transferred to a 10 mL volumetric flask and diluted with ultrapure water, then diluted again 1:10 with ultrapure water.

The concentrations of elements in the leaching solutions were measured by Inductively Coupled Plasma-Mass spectrometry (ICP–MS Agilent Technologies, 7500 cx series) by using the following operating conditions: power 1550 W, carrier gas 1.03 L/min, make-up gas 0.00 L/min, sample depth 8 mm, nebulizer pump 0.1 r.p.s. and spray chamber temperature 2 °C. In order to overcome most of the polyatomic interference by the collision cell, the ICP instrument operated in He mode. A typical performance test in He mode was as follows: He flux 3.1 mL/min, solution containing 10 ppb of 9Be (9000 cps), 45Sc (30,000 cps), 56ArO (300 cps), 115In (30,000 cps), 140Ce (40,000 cps) and 209Bi (12,000 cps). The same solution without 9Be was used as internal standard for ICP–MS measurements. Calibration curves for the investigated elements were obtained using aqueous (1% nitric acid) standard solutions prepared with appropriate dilution of stock standards (Fluka Analytical, Sigma Aldrich). Element concentrations were quantified using ChemStation System Software for ICP–MS (version B.03.07, Agilent Technologies, Inc. 2008, Tokyo, Japan). Limit of detection (LOD) was calculated both as three times the standard deviation (SD) of the element concentration in the calibration blanks, and the ICP–MS raw counts were monitored to assure that signal/noise ratios were >3 (data not shown).

### 2.9. Determination of Cytotoxicity and Antioxidant Activity in Cells

#### 2.9.1. Cell Cultures

Human keratinocyte cell line, HaCaT, were routinely grown in Dulbecco’s modified Eagle’s Medium supplemented with 10% fetal bovine serum, 2 mM L-glutamine, 50 U/mL penicillin, and 50 µg/mL streptomycin at 37 °C in a humidified incubator with 5% CO_2_. To evaluate cytotoxicity and intracellular ROS formation, HaCaT cells were seeded in 96-well plates at 2 × 104 cells/well. All experiments were performed after 24 h of incubation at 37 °C in 5% CO_2_.

#### 2.9.2. Cytotoxicity

Cell viability was evaluated by the reduction of 3-(4,5-dimethyl-2-thiazolyl)-2,5-diphenyl-2H-tetrazolium bromide (MTT) to its insoluble formazan, as previously described [33]. Briefly, HaCaT cells were treated for 24 h with different concentrations of spent coffee extract (0.003–3 mg/mL) at 37 °C in 5% CO2. Subsequently, the treatment medium was replaced with MTT in Hank’s Balanced Salt Solution (HBSS) (0.5 mg/mL) for 2 h at 37 °C in 5% CO2. After washing with HBSS, formazan crystals were dissolved in isopropanol. The amount of formazan was measured (570 nm, reference filter 690 nm) using the multilabel plate reader VICTOR™ X3 (PerkinElmer, Waltham, MA, USA). The cell viability was expressed as percentage of control cells and calculated by the formula: (absorbance of treated neurons/absorbance of untreated neurons) × 100.

#### 2.9.3. Intracellular Reactive Oxygen Species (ROS) Formation

ROS formation was evaluated by fluorescent probe 2′–7′ dichlorodihydrofluoresce in diacetate (H2DCF-DA), as previously described [34]. HaCaT cells were treated for 2 h with different concentrations of spent coffee extract (0.003–0.3 mg/mL) at 37 °C in 5% CO_2_. Subsequently, treatment medium was removed and 100 µL of H2DCF-DA (10 µg/mL) was added to each well for. After 30 min of incubation at room temperature, H2DCF-DA solution was replaced with a solution of H_2_O_2_ (100 µM) for 30 min. The ROS formation was measured (excitation at 485 nm and emission at 535 nm) using the multilabel plate reader VICTOR™ X3 (PerkinElmer). Data are expressed as increased percentage of ROS formation versus untreated cells.

### 2.10. Statistical Analysis

The design experiments, all data processing, along with plots and contour surfaces here presented, were carried out using Nemrod-W^®^ software developed by NemrodW SAS (Marseille, France).

The biological data are shown as mean ± standard error (SEM) of at least three independent experiments. In this regard, statistical analysis was performed using one-way ANOVA (Analysis of Variance) with Dunnett or Bonferroni post hoc test, as appropriate. Analyses were performed using GraphPad PRISM software (version 5.0; GraphPad Software, La Jolla, CA, USA) on a Windows platform.

## 3. Results

### 3.1. Optimization of the Extraction from Spent Coffee Grounds Using the Desirability Approach

On the basis of preliminary results (not reported here), obtained from a previous HPLC–VWD based screening study on the extraction process from spent coffee, the most significant factors identified were the solvent extraction volume and temperature. Consequently, for the process optimization, a quadratic domain was experimentally investigated with a Central Composite Design to estimate the second-order polynomial model of Equation (1), using the Response Surface Design methodology.

The results of ANOVA for the model of the three variable responses under study—the contents of caffeine (*Y*_1_), trigonelline (*Y*_2_), and nicotinic acid (*Y*_3_)—are listed in Table 4. In particular, the Lack of Fit (LOF) test results are reported here to assess whether the models accurately fit the measured experimental data. The LOF test procedure consists in portioning the error (residual sum of squares—*SS_E_*) into a component due to “pure” error (pure error sum of squares—*SS_PE_*), computed on the basis of replicated design points, and a component due to lack of fit (lack of fit sum of squares—*SS_LOF_*). If the *F*-ratio for lack of fit is not significant, it is possible to accept the hypothesis that the model adequately describes the data. In this case, for all three response variables under study, there is no evidence that the quadratic model does not fit the data. The goodness of fit of the proposed model can be also positively assessed according to the values of the coefficients of determination *R*-squared (*R*_2_) and *R*-squared adjusted (*R*_2*A*_) for the degree of freedom (d.f.), which were all higher than 0.90.

The estimated coefficients of the second-order polynomial model of Equation (1) for the three response variables are listed in Table 5. The experimental and predicted values for the contents of caffeine, trigonelline, and nicotinic acid are instead given in Table 5, along with their residuals.

As previously mentioned, for the validation of the postulated model, in the experimental design, three test points (runs 11, 12, and 13) were also included in order to be able to assess whether the model adequately represents the response variables in the experimental domain of interest. The two replicated experiments at the center of the experimental domain (runs 9 and 10) allow one to have an estimation of the repeatability and hence of the size of the experimental error that can be compared to the model residual given by the difference between the observed value of the dependent variable, *Y*i,exp, and the predicted value *Y*i,calc obtained at the test points. In this case, the estimated experimental repeatability values for the three variable responses were: ±9.1648 for *Y*_1_, ±4.3707 for *Y*_2_, and ±0.0424 for *Y*_3_. It was therefore possible to validate the models, since all the differences between the values calculated by the models for the test points and the experimental values measured in the same points were of the order of magnitude of the repeatability of the replicate points at the center of the experimental domain.

Once the models were validated, the experimental data obtained at the check points may be included for a new estimation of the model coefficients to improve the fit. In Table 6, the model coefficients and their significance level estimated with the complete set of the 13 data values are listed. Table 7 reports the new ANOVA table.

To identify an optimal compromise area where the highest response for all the three variables under study are obtained, the desirability approach was considered [25]. This method is developed in two steps:

Step (1) transformation of every response (*Y_i_*) as a function of the objectives under the form of an individual or elementary desirability function (di) by the definition of specific target values, *T_i_*
*d_i_* = *Ti* (*Y_i_*, objectives)

The desirability function assigns a number between 0 and 1 to the possible values of *Y_i_*, where 0 identifies a completely undesirable value of *Y_i_* and 1 the optimal response value.

Step (2) determination of the overall desirability function (*D*), giving the global optimal compromise obtained computing the geometric mean (*G*) of the desirability functions
*D* = *G* (*d*_1_, *d*_2_,…, *d_m_*)

The compromise solution is “favorable” with increasing positive of *D* and becomes “perfect” when *D* = 100%.

In Figure 2, the three individual desirability functions, one for each of the response variables to optimize, are graphically represented.

In order to determine which is the zone, inside the experimental domain of interest, where the estimation of all the responses can be calculated with a probability ≤α% of not respecting predefined constraints for each response, the following constraints were given (Equation (2)):Prob (Caffeine content > 33.89) ≥ 0.95
Prob (Trigonelline content > 7.07) ≥ 0.95(2)
Prob (Nicotinic acid content > 0.08) ≥ 0.95

In this way, it will be possible to assess the reliability of the desirability function, or in other terms, where, inside the optimal zone, it is possible to find solutions respecting the given constraints with a probability ≥(1 − α).

Figure 3 shows the bidimensional contour plot of the overall desirability function respecting the given constraints (Equation (2)) with a probability ≥95%. Here, the variations in *D* values in the experimental domain of interest are graphically represented through the response surface model by determining operating parameter ranges satisfying the predefined objectives and constraints.

In Figure 2, the “design space” is also identified, in other words, the experimental domain where the constraints of Equation (2) are satisfied, given by the following operating parameter ranges:16.1 < Solvent extraction volume < 23.9
65.7 < Extraction Temperature < 107.2

### 3.2. Evaluation of Total Phenol Content and Antioxidant Capacity of Spent Coffee Extracts

According to the results obtained with the desirability approach, an extract was produced in a volume of 20 mL of water and at a temperature of 80 °C for 30 min. In Table 8, the Total Phenol Content (TPC) and the antioxidant capacities (AC) (ABTS, FRAP, and DPPH) of spent coffee extract are reported. It must be noted that TPC and AC may depend not only on caffeic and nicotinic acids but also on other antioxidant molecules that could be present in the extract but that have not been quantified in the present study. If one compares the results of the present study with those reported in the literature, it is possible to note that TPC and AC are frequently higher than some reported in previous studies (Table 1) [6,9,14,15,19]. Among the water extracts, the highest value was obtained under subcritical water extraction at 179 °C for 36 min (TPC: 88.34 mg GAE/g, ABTS: 886.50 TEAC µmol TE/g, DPPH: 382.80 TEAC µmol TE/g) [14]. Autohydrolysis carried out in 15 mL water/g of SCG at 200 °C for 50 min provided the worst results with respect to subcritical water extraction. The results of our study are intermediate and just lower than those obtained under subcritical water extraction at 179 °C for 36 min.

Thus, results of the present study are highly encouraging because it was possible to select milder extraction conditions, which means lower temperature (80 °C) per lower amount of SCG (only 1 g in 20 mL of water as solvent extraction), for a short extraction time (only 30 min).

### 3.3. Determination of Metals and Minerals by Inductively Coupled Plasma Mass Spectrometry (ICP–MS)

According to the experimental method, the concentration of elements contained in the spent coffee extract is presented in Table 9, while in Figure 4 the distributions of elements are graphically expressed.

Generally, elements content in coffee is about 5% m/m [35], and the element types and concentration gives useful information about the origin of growing soils for coffee plants’ cultivation and environmental conditions. These data therefore represent an indication of the coffee authenticity [7,36,37] and are very important when spent coffee wastes are used as feedstock. Potassium is the most abundant element, followed by phosphorus and magnesium [37]. Total mineral extracted with hot water during coffee preparation varies from 0.82% to 3.52%, and potassium ranges from 3.12 to 21.88 mg/g [7,38]. In the present study, the results are in accord with those obtained in coffee revealing that total mineral extract from spent coffee is about 0.81% with about 6 mg/g of potassium that is the most abundant element. The trend of major constituents (mg/g) in the extract was: K >> P > Mg > Ca >Na > S. In the group of minor constituents, elements are in the range from about 0.1 to 8 μg/g while, in the group of trace elements, in which heavy metals are included, the range is from about 0.7 to 40 ng/g demonstrating very low concentration of toxic metals.

### 3.4. In Vitro Results

We initially evaluated the cytotoxicity of spent coffee extract in keratinocyte HaCaT cells. The cells were treated with extract concentrations ranging from 0.003 to 3 mg/mL for 24 h, and cell viability was evaluated by MTT assay. The treatment of HaCaT cells with extract at the concentrations lower than 3 mg/mL did not affect cell viability (Figure 5). The range of concentrations 0.003–0.3 mg/mL was therefore selected to evaluate the ability of extract to counteract the intracellular ROS formation elicited by H_2_O_2_ (100 µM) in HaCaT using the probe H2DCF-DA. As shown in Figure 6, the treatment of HaCaT cells with extract significantly reduced the intracellular ROS formation at 0.03 and 0.3 mg/mL in a dose-dependent manner. Taken together, these results show the ability of spent coffee extract to counteract the ROS formation at intracellular level in keratinocytes cells.

## 4. Conclusions

This study encourages the projects aimed to exploit waste from food, such as spent coffee grounds. This waste, which commonly ends up in the dustbin, could be collected to produce an extract with added values. In particular, the results described here show the advantages resulting from a multiple-response optimization approach based on RSM combined with the desirability function method. In this case, by considering the promising results obtained in terms of phenol content compounds, the total antioxidant capacity, the content in elements, and the in vitro tests, we may consider using spent coffee extract as a source of active attractive ingredient in cosmetic formulations for aged skin. Further studies will be devoted to including the spent coffee ground extract in cosmetic formulations.

## Figures and Tables

**Figure 1 antioxidants-09-00370-f001:**
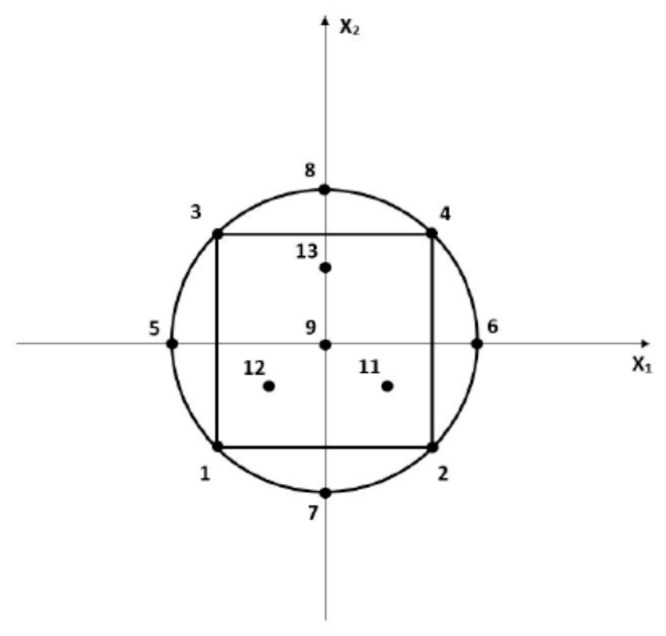
Graphical representation of the Central Composite Design (CCD) used for the optimization study for two variables with three check points (11–13 points) and one replicate point at the center of the experimental design (runs 9 and 10).

**Figure 2 antioxidants-09-00370-f002:**
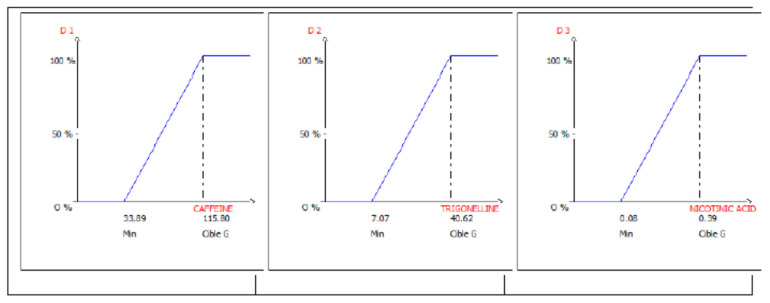
The elementary desirability functions (*d*_1_, *d*_2_ and *d*_3_) for the maximization of the three response variables *Y*_1_ = caffeine, *Y*_2_ = trigonelline, and *Y*_3_ = nicotinic acid.

**Figure 3 antioxidants-09-00370-f003:**
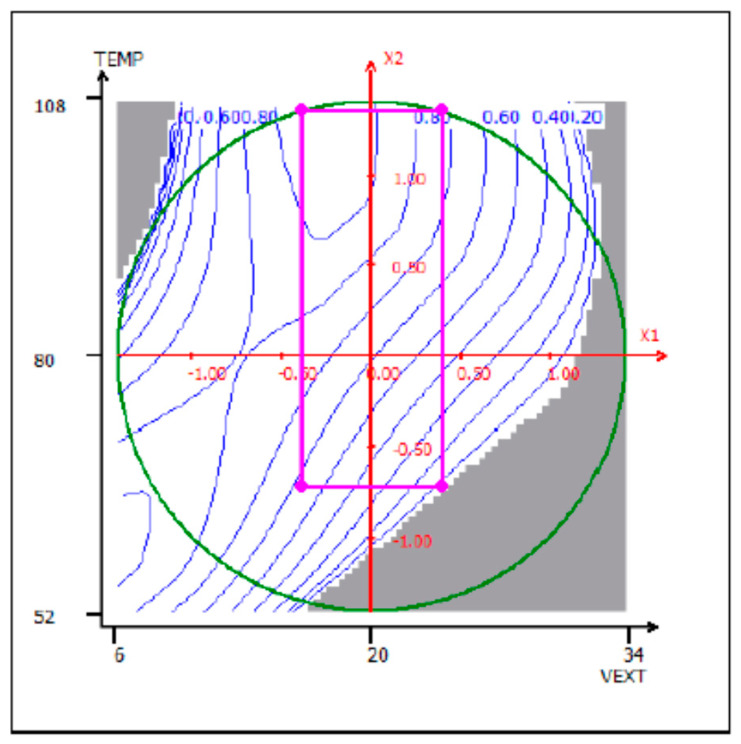
Contour plot of the global desirability function and the design space given by the operating parameter ranges satisfying the constraints of Equation (2).

**Figure 4 antioxidants-09-00370-f004:**
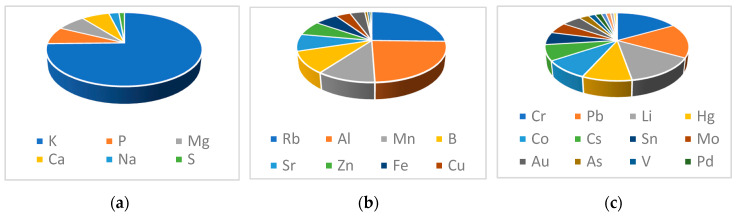
Elements distribution: major elements (**a**), minor elements (**b**) and trace elements (**c**).

**Figure 5 antioxidants-09-00370-f005:**
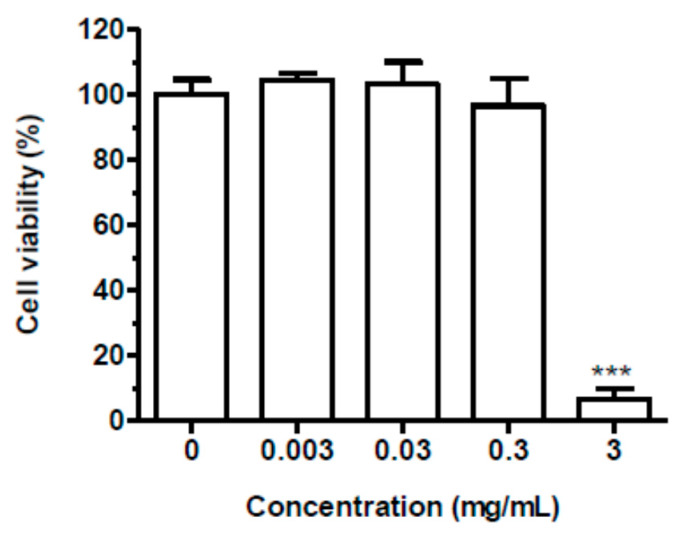
Cytotoxicity of spent coffee extract in HaCaT cells. Cells were treated with various concentrations of spent coffee extract (0.003–3 mg/mL) for 24 h. At the end of treatment, cell viability was evaluated by MTT assay, as described in the method section. Data are expressed as a percentage of control cells and expressed as mean ± SEM of three independent experiments (*** *p* < 0.001 vs. cells untreated; one-way ANOVA with Dunnett post hoc test).

**Figure 6 antioxidants-09-00370-f006:**
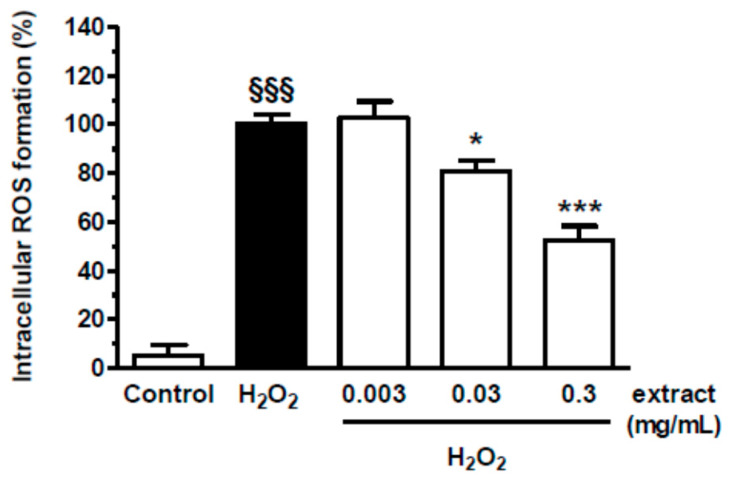
Antioxidant activity of spent coffee extract against the ROS formation induced by H_2_O_2_ in HaCaT cells. Cells were treated with various concentrations of spent coffee extract (0.003–0.3 mg/mL) for 2 h and then treated with H_2_O_2_ (100 μM) for 30 min. At the end of treatment, intracellular ROS formation was evaluated using the fluorescent probe H2DCF-DA, as described in the method section. Data are expressed as increased percentage of ROS formation versus untreated cells and reported as mean ± SEM of three independent experiments (^§§§^
*p* < 0.001 vs. cells untreated; * *p* < 0.05, and *** *p* < 0.001 vs. cells treated with H_2_O_2_; one-way ANOVA with Dunnett or Bonferroni post hoc test).

**Table 2 antioxidants-09-00370-t002:** Process variables, along with the corresponding tested levels (lower and upper level) and the three response variables under study for the extraction optimization from spent coffee.

Process Variables		Coded Variable	Original Units	Coded Units	Response Variables	
*U* _1_	Volume (mL)	*X* _1_	10 30	–1 +1	*Y* _1_	Caffein content
*U* _2_	Temperature (°C)	*X* _2_	60 100	–1 +1
					*Y* _2_ *Y* _3_	Trigonelline content Nicotinic acid content

**Table 3 antioxidants-09-00370-t003:** The Central Composite Design for spent coffee extraction optimization: coded and natural variables under study. Extraction was carried out in water.

Experiment Number	Design (Coded Variables)		Run Order	Plan (Natural Variables)	
				Volume	Temperature
	*X* _1_	*X* _2_		(mL)	(°C)
1	−1.00	−1.00	13	10	60
2	+1.00	−1.00	11	30	60
3	−1.00	+1.00	6	10	100
4	+1.00	+1.00	12	30	100
5	−1.414	0.00	8	6	80
6	+1.414	0.00	9	34	80
7	0.00	−1.414	4	20	52
8	0.00	+1.414	2	20	108
9	0.00	0.00	7	20	80
10	0.00	0.00	3	20	80
11	−0.61	−0.35	1	14	73
12	+0.61	−0.35	10	26	73
13	0.00	+0.70	5	20	94

**Table 4 antioxidants-09-00370-t004:** ANOVA for the quadratic experimental models for caffeine, trigonelline, and nicotinic acid, along with the significant values, where the symbol * stands for a *p*-value (*p*) ≤ 0.05, ** for *p* ≤ 0.01, and *** for *p* ≤ 0.001.

		Caffeine				Trigonelline				Nicotinic Acid			
Source	*d.f.*	Sum of Squares	Mean Square	*F*-ratio	Sig.	Sum of Squares	Mean Square	*F*-ratio	Sig.	Sum of Squares	Mean Square	*F*-ratio	Sig.
Regression	5	2.70279 × 10^4^	5.40557 × 10^3^	38.498	0.18 **	5.09583 × 10^3^	1.01916 × 10^3^	42.163	0.15 **	0.2950	0.0590	17.47	0.80 **
Residual *SS_E_*	4	5.61648 × 10^2^	1.40412 × 10^2^			9.66891 × 10^1^	2.41723 × 10^1^			0.0135	0.0034		
Lack of fit *SS_LOF_*	3	4.77654 × 10^2^	1.59218 × 10^2^	1.896	48.0	7.75867 × 10^1^	2.58622 × 10^1^	1.354	54.7	0.0117	0.0039	2.169	45.4
Error *SS_PE_*	1	8.39938 × 10^1^	8.39938 × 10^1^			1.91024 × 10^1^	1.91024 × 10^1^			0.0018	0.0018		
Total *SS*	9	2.75895 × 10^4^				5.19252 × 10^3^				0.3085			

**Table 5 antioxidants-09-00370-t005:** Estimated coefficients of the second-order polynomial model of Equation 1 for the three response variables, along with the significant values, where the symbol * stands for a *p*-value (*p*) ≤ 0.05, ** for *p* ≤ 0.01, and *** for *p* ≤ 0.001.

	Model Fitting without Test Points						Model Fitting with the Complete Set of 13 Data Values					
	Caffeine		Trigonelline		Nicotinic Acid		Caffeine		Trigonelline		Nicotinic Acid	
	Coeff.	Sig. %	Coeff.	Sign. %	Coeff.	Sig. %	Coeff.	Sig. %	Coeff.	Sig. %	Coeffic.	Sig. %
*b* _0_	118.98	0.0143 ***	30.33	0.0950 ***	0.39	0.0697 ***	116.89	<0.01 ***	30.89	<0.01 ***	0.37	<0.01 ***
*b* _1_	−54.60	0.0204 ***	−19.70	0.0353 ***	0.01	82.0	−54.24	<0.01 ***	−19.78	<0.01 ***	0.01	75.6
*b* _2_	17.18	1.51 *	14.70	0.109 **	0.10	0.783 **	17.78	0.0888 ***	14.61	<0.01 ***	0.10	0.0338 ***
*b* _11_	−2.28	70.5	6.31	5.4	−0.06	8.1	−1.52	71.1	6.09	0.614 **	−0.05	2.44 *
*b* _22_	−14.35	6.3 *	−0.22	93.0	0.04	25.8	−13.16	1.26 *	−0.51	75.5	0.04	5.2
*b* _12_	4.66	47.5	−4.73	12.7	0.21	0.203 **	4.41	37.8	−4.67	4.09 *	0.21	<0.01 ***

**Table 6 antioxidants-09-00370-t006:** Experimental values (*Y**i,exp*), predicted values (*Y**i,cal*) for the contents of caffeine (*Y*_1_), Trigonelline (*Y*_2_), and Nicotinic acid (*Y*_3_), along with the model residuals.

	Caffeine			Trigonelline			Nicotinic Acid		
N° Exp	*Y*_1_,*exp*	*Y*_1_,*cal*	Difference	*Y*_2_,*exp*	*Y*_2_,*cal*	Difference	*Y*_3_,*exp*	*Y*_3_,*cal*	Difference
1	145.860	144.4302	1.4298	35.692	36.6957	−1.0037	0.5080	0.4616	0.0464
2	35.0080	25.9049	9.1031	10.7200	6.7551	3.9649	0.0760	0.0561	0.0199
3	157.822	169.4687	−11.6467	70.9080	75.5516	−4.6436	0.3010	0.2500	0.0510
4	65.6290	69.6024	−3.9734	27.0300	26.7050	0.3250	0.7000	0.6756	0.0244
5	197.722	190.9433	6.7787	74.1740	70.2787	3.8953	0.2010	0.2561	−0.0551
6	33.8860	38.0692	−4.1832	11.9250	15.1277	−3.2027	0.2530	0.2702	−0.0172
7	58.7620	66.8046	−8.0426	7.0710	9.3247	−2.2537	0.2840	0.3169	−0.0329
8	125.558	114.9199	10.6381	53.4350	50.4888	2.9462	0.5630	0.6024	−0.0394
9	125.414	118.9854	6.4286	33.4090	30.3324	3.0766	0.4200	0.3886	0.0314
10	112.453	118.9854	−6.5324	27.2280	30.3324	−3.1044	0.3600	0.3886	−0.0286
11	134.492	144.1289	−8.6369	40.1900	38.2579	1.9321	0.3400	0.3749	−0.0449
12	73.3240	76.6523	−3.3283	17.3240	38.2579	0.7171	0.2600	0.2937	−0.0437
13	125.544	123.9834	1.5606	40.6470	40.5170	0.1300	0.4520	0.4777	−0.0257

**Table 7 antioxidants-09-00370-t007:** ANOVA for the quadratic experimental models for caffeine, trigonelline, and nicotinic acid estimated on the complete experimental data set.

		Caffeine				Trigonelline				Nicotinic Acid			
Source	*d.f.*	Sum of Squares	Mean Square	*F*-ratio	Sig.	Sum of Squares	Mean Square	*F*-ratio	Sig.	Sum of Squares	Mean Square	*F*-ratio	Sig.
Regression	5	2.92657 × 10^4^	5.85315 × 10^3^	65.297	<0.01 ***	5.46235 × 10^3^	1.09247× 10^3^	76.80	<0.01 ***	0.3126	0.0625	28.92	0.00 ***
Residual *SS_E_*	7	6.27473 × 10^2^	8.96391 × 10^1^			9.95740 × 10^1^	1.42248× 10^1^			0.0151	0.0022		
Lack of fit *SS_LOF_*	6	5.43480 × 10^2^	9.05800 × 10^1^	1.078	62.7	8.04716 × 10^1^	1.34119× 10^1^	0.702	72.2	0.0133	0.0022	1.235	59.7
Error *SS_PE_*	1	8.39938 × 10^1^	8.39938 × 10^1^			1.91024 × 10^1^	1.91024× 10^1^			0.0018	0.0018		
Total *SS*	12	2.98932 × 10^4^				5.56192 × 10^3^				0.3277			

**Table 8 antioxidants-09-00370-t008:** Total Antioxidant Capacity (TAC) and antioxidant activities determined according to the FRAP, ABTS, and DPPH methods of spent coffee extract obtained using the desirability function approach.

TPC mg GAE/g	FRAP TEAC (µmol TE/g)	ABTS TEAC (µmol TE/g)	DPPH TEAC (µmol TE/g)
61.49 ± 1.36	311.62 ± 22.65	735.47 ± 0.60	324.51 ± 13.58

**Table 9 antioxidants-09-00370-t009:** Elemental constituents in the extract referred to 1 g of initial spent coffee.

	mg/g	ELEMENTS	µg/g	ELEMENTS	ng/g
**K**	6.03	**Rb**	7.98	**Cr**	38.86
**P**	0.62	**Al**	7.56	**Pb**	38.10
**Mg**	0.57	**Mn**	3.35	**Li**	32.97
**Ca**	0.55	**B**	3.15	**Hg**	21.76
**Na**	0.20	**Sr**	2.45	**Co**	20.71
**S**	0.11	**Zn**	2.13	**Cs**	17.39
		**Fe**	1.85	**Sn**	14.06
		**Cu**	1.20	**Mo**	12.54
		**Ba**	1.19	**Au**	11.88
		**Ni**	0.23	**As**	6.08
		**Ga**	0.19	**V**	4.28
		**Ti**	0.15	**Pd**	3.90
				**Ag**	2.95
				**Cd**	2.85
				**Tl**	2.00
				**Sb**	1.43
				**U**	0.76

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
