# Peer review of "Optimization of the Extraction from Spent Coffee Grounds Using the Desirability Approach"

_antioxidants, 2020, doi:10.3390/antiox9050370_

Round 1

Reviewer 1 Report

See attached file

Reviewer 2 Report

This paper refers to valorization of spent coffe grounds, and extractio techniques of added value compounds to be used in cosmetics. This is a valid approach, but the authors refer to a desireability approch, which could be further explained, as it is not very clear what the novelty of this approach is throuhout the text.

  • The introduction is very brief and would benefit from including the extraction yields of spent coffee grounds from other techniques, and which types of compounds are extarcted from each one. The introduction should include the novelty and advanages brought by this desirability approach.
  • Throughout the text, the superscript and subcript of chemical formulas and units should be corrected
  • line 115- "that was"...the  value is missing.
  • line 245- although permilimanery results are not reported, at least a refernce value should be given
  • line 349- authors say that the antioxaidant values obtained are in the same order of magnitude but table 7 shows values in umol/g and the refernce values from reported works are in mmol/g. This should be refered in the text or corrected.
  • The conclusions are a bit vague and brief. The authiors should emphasize the adavantages of using all the matehmatical treatment  performed, and how this would be useful in the future.

Round 2

Reviewer 1 Report

the correction and the advice, I've seen, that has been included. I've not to hint anything else.